Cellular heterogeneity map of diverse immune and stromal phenotypes within breast tumor microenvironment

Li Yuan liyuanpumc@gmail.com 1
Chen Zuhua 2
Wu Long 1
Ye Junjie 1
Tao Weiping taowpwp@sina.com 1
1 Department of Oncology, Renmin Hospital of Wuhan University , Wuhan , China
2 Department of Oncology, Tongji Hospital, Tongji Medical College, Huazhong University of Science and Technology , Wuhan , China
Gomez Shawn
Electronic publication date: 2020 Jul 10
Publication date: 2020
Volume: 8
Electronic Location ID: e9478
Received 2020 Jan 19; Accepted 2020 Jun 13
Copyright: ©2020 Li et al.
Copyright year: 2020
Copyright holder: Li et al.
License: This is an open access article distributed under the terms of the Creative Commons Attribution License, which permits unrestricted use, distribution, reproduction and adaptation in any medium and for any purpose provided that it is properly attributed. For attribution, the original author(s), title, publication source (PeerJ) and either DOI or URL of the article must be cited.
License URL: https://creativecommons.org/licenses/by/4.0/

Keywords: Breast cancer, Immune, Stromal, Cellular heterogeneity

Funding: National Natural Science Foundation of China 81902369 This work was supported by National Natural Science Foundation of China under Grant 81902369. The funders had no role in study design, data collection and analysis, decision to publish, or preparation of the manuscript.

==============================
Background

Cellular heterogeneity within the tumor microenvironment is essential to tumorigenesis and tumor development. A high-resolution global view of the tumor-infiltrating immune and stromal cells in breast tumors is needed.

Methods

xCell was used to create a cellular heterogeneity map of 64 cell types in 1,092 breast tumor and adjacent normal tissues. xCell digitally dissects tissue cellular heterogeneity based on gene expression. Integrated statistical analyses were then performed.

Results

There were noticeable differences between the cell fractions in tumor tissues and normal tissues. Tumors displayed higher proportions of immune cells, including CD4+ Tem, CD8+ naïve T cells, and CD8+ Tcm compared with normal tissues. Immune inhibitory receptors (PD1, CTLA4, LAG3 and TIM3) were co-expressed on certain subtypes of T cells in breast tumors, and PD1 and CTLA4 were both positively correlated with CD8+ Tcm and CD8+ T cells. 28 cell types were significantly associated with overall survival in univariate analysis. CD4+ Tem, CD8+ Tcm, CD8+ T-cells, CD8+ naive T-cells, and B cells were positive prognostic factors but CD4+ naive T-cells were negative prognostic factors for breast cancer patients. TDRD6 and TTK are promising T cell and B cell targets for tumor vaccines. Endothelial cells and fibroblasts were significantly less prevalent in tumor tissues; astrocytes and mesangial cells were negatively correlated with the T stage. Mesangial cells and keratinocytes were found to be favorable prognostic factors and myocytes were negative prognostic factors. Five cell types were found to be independent prognostic factors and we used these to create a reliable prognostic model for breast cancer patients. Cellular heterogeneity was discovered among different breast cancer subtypes by Her2, ER, and PR status. Tri-negative patients had the highest fraction of immune cells while luminal type patients had the lowest. The various cells may have diverse or opposing roles in the prognosis of breast cancer patients.

Conclusions

We created a uniquecellular map for the diverse heterogeneity of immune and stromal phenotypes within the breast tumor microenvironment. This map may lead to potential therapeutic targets and biomarkers with prognostic utility.

Introduction

Breast cancer is a common cancer in women and drug resistance and distal metastasis remain major causes of mortality despite improvements in the early diagnosis and treatment of the disease (Cassetta & Pollard, 2017). Tumors have complex microenvironments composed of malignant cells, immune cells, and stromal infiltrate. Growing evidence suggests that this microenvironment plays a fundamental role in the development of malignancy and resistance to therapy (Noy & Pollard, 2014). Tumor-infiltrating cells can demonstrate either tumor-suppressing or tumor-promoting effects, depending on the cancer type. For instance, regulatory T cells (Tregs) and tumor associated macrophages (TAMs) are associated with pro-tumor functions (De Palma & Lewis, 2013; Nishikawa & Sakaguchi, 2014; Noy & Pollard, 2014), but CD8+ T cells are associated with improved clinical outcomes and better responses to immunotherapy (Tumeh et al., 2014). Research in cancer immunology has led to the development and approval of checkpoint blockers. These remarkably effective drugs augment T cell activity by blocking cytotoxic lymphocyte antigen-4 (CTLA4), programmed cell death protein 1 (PD1), and PD1 ligand (PDL1).

A better understanding of the cellular heterogeneity within the tumor microenvironment may reveal predictive biomarkers, improve existing treatments, and help to develop novel therapeutic strategies. Cellular heterogeneity is traditionally determined using flow cytometry and immunohistochemistry, however, these methods are extremely difficult to apply to solid tumors with limited throughput (Gentles et al., 2015). Bioinformatics advancements have created novel methods that dissect cellular heterogeneity based on gene expression profiles (Abbas et al., 2009; Newman et al., 2015; Rooney et al., 2015; Shen-Orr & Gaujoux, 2013). For instance, CIBERSORT can estimate the abundances of 22 immune cell types (Newman et al., 2015). However, rare subsets of immune cells and stromal cells recognized to be important in the promotion or inhibition of tumor growth, invasion, and metastasis are ignored by CIBERSORT (Galon et al., 2006; Hanahan & Coussens, 2012). xCell can define 64 cell types within tissues, including immune and stroma cells (Aran, Hu & Butte, 2017a; Aran, Hu & Butte, 2017b). Recent analysis with xCell reveals that plasma cells and CD4+ Tcm in the tumor microenvironment may play a role in the progression of triple-negative breast cancer (Deng et al., 2019), although only immune cells were investigated. We used xCell to digitally depict the cellular heterogeneity map within the breast tumor microenvironment to reveal potential interactions and to uncover predictive biomarkers or therapeutic targets for breast cancer.

Methods and Materials

Data curation and cohort characteristics

The RNA-seq data and clinical parameters from 1,092 patients with breast cancer were obtained from The Cancer Genome Atlas (TCGA) data portal (https://portal.gdc.cancer.gov/). Of the 1,092 breast cancer patients included in this study, 112 paired normal tissues were identified and recurrent tumor tissues were excluded. The clinical characteristics of the cohort are listed in Table 1. 276 known Cancer/Testis (CT) genes were downloaded from CTDatabase (http://www.cta.lncc.br/).

Table 1 Clinical characteristics of the 1,092 breast cancer patients from TCGA.

Characteristic	TCGA BRCA (N = 1,092)	
Age median (range)	59 (26–90)	
Sex		
Female	1,080 (98.9%)	
Male	12 (1.1%)	
Tumor stage		
T1	279 (25.5%)	
T2	635 (58.2%)	
T3	138 (12.6%)	
T4	40 (3.7%)	
Lymph node stage		
N (-)	333 (30.5%)	
N (+)	759 (69.5%)	
Metastasis stage		
M (-)	903(82.7%)	
M (+)	189(17.3%)	
TNM stage		
TNM I	181 (16.5%)	
TNM II	637 (58.3%)	
TNM III	254 (23.3%)	
TNM IV	20 (18.3%)	
Subtypes		
Luminal	426 (39.0%)	
Her2+_HR+	59 (5.4%)	
Her2+_HR-	30 (2.7%)	
Tri-negative	97 (8.9%)	
Unknown	480(44.0%)	
Status		
Living	940 (86.1%)	
Decreased	152 (13.9%)	

Bioinformatics analysis

xCell (http://xcell.ucsf.edu/) is a high-resolution gene-signature-based method for cell type enrichment for up to 64 cell types, including immune and stroma cells. We used xCell R package (Aran, Hu & Butte, 2017a) (Beta version) from GitHub in R (version 3.3.1) to deconvolute the cellular heterogeneity within the breast tumor microenvironment from RNA sequencing data. We determined the cellular heterogeneity of 1,092 breast tumor tissues and 112 normal tissues using the xCell method. The 64 cell types were divided into four groups, including 34 immune cells, 13 stromal cells, 9 stem cells, and 8 other cells (Table 2). Over half of the 64 cell types were immune cells, providing a full view of the innate and adaptive immune status with detailed cell subtypes, including CD4+ naive cells, CD4+ T-cells, CD4+ Tcm, and CD4+ Tem. Stromal cells, including fibroblasts, osteoblasts, and pericytes were also included. ImmuneScore and StromaScore were generated by the xCell package using the sums of fractions of certain cell types (Aran, Hu & Butte, 2017b). Tests for differences and correlations were performed. We used the t-Distributed Stochastic Neighbor Embedding (t-SNE) method with tsne package (version 0.1-3) to perform cluster analysis based on cell fraction types.

Survival analysis

Univariate and multivariate COX regressions were performed using the survival package (version 3.1-7) to search for survival-associated genes. The best cutoff value for each factor was determined using the Survminer package (version 0.4.6). Significant prognostic factors were displayed in a forest plot and the most significant factors were further evaluated using multivariate analysis. The final prognostic model was built with five independent factors including CD8+ T cells, mesangial cells, NKT, keratinocytes, and class-switched memory B cells. We used the survivalROC package (Version 1.0.3) in R, which uses a time-dependent ROC curve estimation with censored data (Heagerty, Lumley & Pepe, 2000) to compare the aptitude of the individual prognostic factors. The final prognostic model was used to generate the area under the curve (AUC) of the receiver-operator characteristic (ROC) curve for each parameter.

Table 2 Abbreviations and statistical summary of enriched fractions of the Sixty-four cell types.

Full name	Abbreviations	Group	TumorN = 1, 092(Mean ± SD)	NormalN = 112(Mean ± SD)	Paired TumorN = 112(Mean ± SD)	P value	P value(paired)	
Activated dendritic cells	aDC	Immune	0.116 ± 0.110	0.016 ± 0.045	0.103 ± 0.104	<0.001	<0.001	
B-cells	/	Immune	0.031 ± 0.085	0.004 ± 0.039	0.021 ± 0.052	<0.001	<0.001	
Basophils	/	Immune	0.078 ± 0.056	0.058 ± 0.040	0.071 ± 0.054	<0.001	0.077	
CD4+ naive T-cells	/	Immune	0.010 ± 0.029	0.003 ± 0.015	0.008 ± 0.020	0.009	0.004	
CD4+ T-cells	/	Immune	0.003 ± 0.010	0.000 ± 0.003	0.001 ± 0.004	0.101	0.036	
Central memory CD4+ T Cell	CD4+ Tcm	Immune	0.009 ± 0.015	0.016 ± 0.019	0.003 ± 0.007	0.006	<0.001	
Effector memory CD4+ T cell	CD4+ Tem	Immune	0.013 ± 0.022	0.000 ± 0.002	0.010 ± 0.018	<0.001	<0.001	
CD4+ memory T-cells		Immune	0.007 ± 0.017	0.000 ± 0.004	0.007 ± 0.012	<0.001	<0.001	
CD8+ naive T-cells	/	Immune	0.012 ± 0.011	0.005 ± 0.007	0.011 ± 0.010	<0.001	<0.001	
CD8+ T-cells	/	Immune	0.018 ± 0.039	0.006 ± 0.016	0.016 ± 0.032	<0.001	0.001	
Central memory CD8+ T Cell	CD8+ Tcm	Immune	0.026 ± 0.049	0.005 ± 0.015	0.027 ± 0.046	<0.001	<0.001	
Effector memory CD8+ T cell	CD8+ Tem	Immune	0.001 ± 0.007	0.000 ± 0.000	0.000 ± 0.002	0.805	0.608	
Conventional dendritic cells	cDC	Immune	0.037 ± 0.046	0.072 ± 0.057	0.040 ± 0.043	<0.001	<0.001	
Class-switched memory B-cells	/	Immune	0.026 ± 0.031	0.002 ± 0.014	0.020 ± 0.023	<0.001	<0.001	
Dendritic cells	DC	Immune	0.005 ± 0.011	0.005 ± 0.011	0.004 ± 0.008	0.826	0.927	
Eosinophils	/	Immune	0.000 ± 0.001	0.000 ± 0.000	0.000 ± 0.001	0.737	0.763	
Immature dendritic cells	iDC	Immune	0.042 ± 0.073	0.167 ± 0.171	0.047 ± 0.053	<0.001	<0.001	
Macrophages	/	Immune	0.038 ± 0.036	0.008 ± 0.023	0.038 ± 0.034	<0.001	<0.001	
Inflammatory (M1) macrophages	Macrophages M1	Immune	0.022 ± 0.027	0.003 ± 0.010	0.019 ± 0.027	<0.001	<0.001	
Reparative (M2) macrophages	Macrophages M2	Immune	0.031 ± 0.021	0.030 ± 0.034	0.031 ± 0.017	0.001	0.003	
Mast cells	/	Immune	0.024 ± 0.012	0.015 ± 0.008	0.025 ± 0.010	<0.001	<0.001	
Memory B-cells	/	Immune	0.006 ± 0.028	0.001 ± 0.011	0.003 ± 0.014	<0.001	0.083	
Monocytes	/	Immune	0.004 ± 0.012	0.003 ± 0.013	0.004 ± 0.011	0.007	0.040	
naive B-cells	/	Immune	0.004 ± 0.016	0.001 ± 0.009	0.003 ± 0.009	<0.001	0.004	
Neutrophils	/	Immune	0.000 ± 0.001	0.003 ± 0.005	0.000 ± 0.000	<0.001	<0.001	
Nature killer cells	NK cells	Immune	0.000 ± 0.002	0.000 ± 0.000	0.000 ± 0.001	0.342	0.759	
Natural killer T cells	NKT	Immune	0.061 ± 0.039	0.028 ± 0.036	0.043 ± 0.028	<0.001	<0.001	
Plasmacytoid dendritic cells	pDC	Immune	0.008 ± 0.019	0.000 ± 0.000	0.007 ± 0.017	<0.001	0.004	
Plasma cells	/	Immune	0.019 ± 0.018	0.001 ± 0.003	0.015 ± 0.012	<0.001	<0.001	
pro B-cells	/	Immune	0.009 ± 0.017	0.000 ± 0.000	0.006 ± 0.013	<0.001	<0.001	
Gamma delta T cells	Tgd cells	Immune	0.003 ± 0.009	0.000 ± 0.000	0.004 ± 0.011	<0.001	0.003	
Regulatory T cells	Tregs	Immune	0.012 ± 0.016	0.004 ± 0.009	0.013 ± 0.016	<0.001	<0.001	
Type 1 T helper (Th1) cells	Th1 cells	Immune	0.133 ± 0.092	0.014 ± 0.026	0.101 ± 0.067	<0.001	<0.001	
Type 2 T helper (Th2) cells	Th2 cells	Immune	0.075 ± 0.096	0.002 ± 0.007	0.082 ± 0.091	<0.001	<0.001	
Astrocytes	/	Others	0.076 ± 0.069	0.110 ± 0.065	0.081 ± 0.064	<0.001	<0.001	
Epithelial cells	/	Others	0.365 ± 0.092	0.277 ± 0.146	0.354 ± 0.089	<0.001	<0.001	
Hepatocytes	/	Others	0.001 ± 0.002	0.005 ± 0.003	0.002 ± 0.002	<0.001	<0.001	
Keratinocytes	/	Others	0.047 ± 0.038	0.065 ± 0.037	0.046 ± 0.038	<0.001	<0.001	
Melanocytes	/	Others	0.010 ± 0.008	0.008 ± 0.008	0.010 ± 0.009	0.011	0.162	
Mesangial cells	/	Others	0.014 ± 0.015	0.034 ± 0.013	0.015 ± 0.015	<0.001	<0.001	
Neurons	/	Others	0.004 ± 0.008	0.004 ± 0.002	0.004 ± 0.006	<0.001	<0.001	
Sebocytes	/	Others	0.016 ± 0.019	0.009 ± 0.009	0.017 ± 0.024	<0.001	<0.001	
Common lymphoid progenitor	CLP	Stem	0.039 ± 0.026	0.013 ± 0.015	0.041 ± 0.026	<0.001	<0.001	
Common myeloid progenitor	CMP	Stem	0.001 ± 0.003	0.003 ± 0.004	0.001 ± 0.003	<0.001	0.008	
Granulocyte-macrophage progenitor	GMP	Stem	0.002 ± 0.006	0.002 ± 0.006	0.002 ± 0.008	0.001	0.025	
Hematopoietic stem cells	HSC	Stem	0.150 ± 0.110	0.492 ± 0.191	0.174 ± 0.125	<0.001	<0.001	
Megakaryocytes	/	Stem	0.004 ± 0.004	0.021 ± 0.009	0.005 ± 0.005	<0.001	<0.001	
Multipotent progenitors	MPP	Stem	0.00 ± 0.001	0.000 ± 0.000	0.000 ± 0.000	<0.001	0.027	
Erythrocytes	/	Stem	0.000 ± 0.000	0.000 ± 0.000	0.000 ± 0.000	0.471	0.306	
Megakaryocyte-erythroid progenitor	MEP	Stem	0.035 ± 0.030	0.011 ± 0.017	0.027 ± 0.022	<0.001	<0.001	
Platelets	/	Stem	0.000 ± 0.002	0.000 ± 0.001	0.000 ± 0.001	0.062	0.334	
Adipocytes	/	Stromal	0.050 ± 0.088	0.382 ± 0.207	0.063 ± 0.102	<0.001	<0.001	
Chondrocytes	/	Stromal	0.035 ± 0.037	0.053 ± 0.022	0.040 ± 0.038	<0.001	<0.001	
Endothelial cells	/	Stromal	0.058 ± 0.056	0.209 ± 0.104	0.066 ± 0.059	<0.001	<0.001	
Fibroblasts	/	Stromal	0.058 ± 0.068	0.172 ± 0.078	0.063 ± 0.074	<0.001	<0.001	
Lymphatic endothelial cells	ly Endothelial cells	Stromal	0.020 ± 0.027	0.110 ± 0.071	0.021 ± 0.025	<0.001	<0.001	
Mesenchymal stem cells	MSC	Stromal	0.281 ± 0.144	0.019 ± 0.054	0.252 ± 0.139	<0.001	<0.001	
Microvascular endothelial cells	mv Endothelial cells	Stromal	0.032 ± 0.032	0.104 ± 0.062	0.029 ± 0.028	<0.001	<0.001	
Myocytes	/	Stromal	0.004 ± 0.010	0.009 ± 0.042	0.005 ± 0.008	0.012	0.387	
Osteoblast	/	Stromal	0.025 ± 0.031	0.005 ± 0.013	0.018 ± 0.024	<0.001	<0.001	
Pericytes	/	Stromal	0.053 ± 0.052	0.027 ± 0.035	0.057 ± 0.055	<0.001	<0.001	
Preadipocytes	/	Stromal	0.024 ± 0.040	0.179 ± 0.090	0.032 ± 0.045	<0.001	<0.001	
Skeletal muscle	/	Stromal	0.001 ± 0.011	0.010 ± 0.101	0.001 ± 0.003	0.442	0.365	
Smooth muscle	/	Stromal	0.133 ± 0.091	0.125 ± 0.067	0.175 ± 0.088	0.345	<0.001	
ImmuneScore	/	/	0.082 ± 0.103	0.029 ± 0.049	0.074 ± 0.079	<0.001	<0.001	
StromaScore	/	/	0.083 ± 0.088	0.382 ± 0.174	0.096 ± 0.097	<0.001	<0.001	

Statistical analysis

Differentially enriched cell types between groups were compared using the Student’s t-test (two groups) or one-way ANOVA analysis (three groups). Correlation analyses were performed using the Spearman method. The survival curves were compared using the Kaplan–Meier method and log-rank test. All tests were two-sided and p < 0.05 was considered to be statistically significant unless otherwise noted. Data were analyzed using R (version 3.4.4).

Results

Breast tumor tissues had higher fractions of immune cells than normal tissues

The median fractions for each cell type were calculated for normal and breast tumor tissues and the proportions of the 64 cells were found to differ between breast tumor and normal tissues (Fig. 1A, Table 2). Breast tumor tissue had higher fraction of immune cells with red to light blue markers whereas normal tissue had larger proportions of stem and stromal cells with blue to red markers (Fig. 1A). Unsupervised cluster analysis revealed that breast tumor tissues and the adjacent normal tissues were clustered into different groups. Immune cells were also clustered into several subgroups (Fig. 1B), indicating that the cellular heterogeneity in tumor vs. normal tissues was much greater than that in a single sample. Dimensionality reduction and visualization by t-Distributed Stochastic Neighbor Embedding (t-SNE) also suggested clear difference between the tumor and normal tissues (Fig. 1C).

Figure 1 Differences of cellular heterogeneity between breast tumor tissue and normal tissues.

(A) Median fractions of Sixty-four cell types in breast tumor and normal tissues. Sixty-four cell types were grouped into four groups: immune, stem, stromal, and other cells. (B) Heatmap of fractions of 64 cell types in 1,092 breast tumor tissues and 112 adjacent normal tissues. (C) Dimensionality reduction and visualization by t-Distributed Stochastic Neighbor Embedding (t-SNE) clustering based on cell fractions. (D) to W. Dot plots of fractions of certain cell types in breast tumor and normal tissues. Lines between dots indicated paired tissues from the same breast cancer patient. *, P < 0.05. **, P < 0.01. ***, P < 0.001.

We compared the fractions of each cell type between breast tumor and normal tissues, revealing dramatic differences in the number of cell types between the tumor and normal tissues (Fig. 1D–1W). Immune cells tended to be more diverse compared to stem or stromal cells. For innate immune cells, neutrophils were more prevalent in normal tissues whereas eosinophils were higher in tumor tissues (Fig. 1D–1E). There was not a significant difference in DC cells between normal and tumor tissues. However, iDC was significantly lower in tumor tissues but pDC and aDC were significantly higher (Fig. 1F–1I). This phenomenon was also seen in macrophages, in which macrophage M1 was higher in tumor tissues while macrophage M2 was lower (Fig. 1J–1K). CD4+ Tcm was found to be significantly lower in tumor tissues, while CD4+ Tem, CD8+ naïve T cells, and CD8+ Tcm were significantly higher (Fig. 1L–1O). Plasma cells, pro B cells, Tgd, Th1, Th2 cells, and Tregs were also found to be significantly higher in tumor tissues (Fig. 1P–1U). Representative stromal cells, such as endothelial cells and fibroblasts, were found to be significantly lower in tumor tissues (Fig. 1V–1W). Differential analysis with paired tumor and normal tissues showed similar patterns (Table 2).

Inhibitory receptors were co-expressed on certain subtypes of T cells

Inhibitory receptors, including PD1, CTLA4, LAG3, and TIM3, expressed on T cells often led to T-cell exhaustion allowing tumors to evade the immune response (Huang et al., 2017; Nirschl & Drake, 2013). The use of specific antibodies to inhibit CTLA4 or PD1 and overcome immune suppression and tumor regression is promising (Brahmer et al., 2012; Callahan, Wolchok & Allison, 2010).

We investigated the correlations between these inhibitory receptors and CD4+/CD8+ T cells. Heatmaps suggested that the expression patterns of the inhibitory receptors correlated with specific subsets of T cells with distinctions between tumor tissues and normal tissues (Fig. 2A and 2B). Correlation analyses also demonstrated that CD8+ T-cells, CD8+ Tcm, CD8+ naive T-cells, CD4+ memory T cells, and CD4+ naïve T cells were all positively correlated with expressions of these inhibitory receptors in tumor tissues, especially with PD1 and CTLA4 (P < 0.05) (Fig. 2C and 2D). CD8+ Tem, CD4+ Tcm, and CD4+ T-cells were not strongly correlated with the expressions of the inhibitory receptors (Fig. 2C). Only a few T cells were significantly correlated with these inhibitory receptors in normal tissues; TIM3 expression was negatively correlated with CD4+ Tcm (Fig. 2C and 2F). We observed a significant correlation among the expression of inhibitory receptors (Fig. 2E and 2G).

Cancer/testis genes TDRD6 and TTK show promise as breast cancer targets

Cancer/Testis (CT) genes are a cluster of tumor-associated proteins normally expressed in germ cells and different cancers. However, they are not typically seen in normal somatic cells (Scanlan et al., 2002). The limited expression of CT genes makes them ideal cancer and immunotherapy biomarkers.

We studied the antitumor immunity response to antigens generated by CT genes by examining 276 known CT genes (obtained from the CTDatabase) for their association with immune components. The significant associations between immune cells and CT genes (P < 0.001) are shown in Fig. 3A. Most of the adaptive immune cells were significantly correlated with CT genes. T cells, such as CD8+ T cells and aDC, which belonged to adaptive and innate immune responses, were positively correlated with most of the CT genes (Fig. 3A–3C). Moreover, two CT genes, TDRD6 and TTK, were positively correlated with a number of immune cells, especially the CD4+/CD8+ T cells (Fig. 3D and 3E), implying strong host immune reactions to these two cancer antigens.

Figure 2 Expression patterns of inhibitory receptors on CD4+/CD8+ T cells.

(A and B) Heatmaps of expression of inhibitory receptors including PD1, CTLA4, LAG3 and TIM3, and fractions of CD4+/CD8+ T cells in tumor tissues (A) and normal tissues (B). Data were transformed by rank and normalized. (C) Clustered correlation matrixes among inhibitory receptors and CD4+/CD8+ T cells in tumor tissues (up-right triangle) and normal tissues (low-left triangle). (D) Dot plot of correlations between PD1 expression and fractions of CD8+ Tcm in tumor tissues. (E) Dot plot of correlations between PD1 expression and CTLA4 expression in tumor tissues. (F) Dot plot of correlations between TIM3 expression and fractions of CD4+ Tcm in normal tissues. (G) Dot plot of correlations between PD1 expression and LAG3 in normal tissues.

Figure 3 Correlations between cancer/testis genes and immune cells.

(A) Significant correlations between cancer/testis (CT) genes and immune cells. Scaled color dots represented significant correlations between CT genes and immune cells (P < 0.001) and red dots represented positive correlations while blue dots represent negative correlations. (B and C) CD8+ naïve T-cells and aDC were positively correlated with most of the CT genes. (D and E) TDRD6 and TTK were positively correlated with a number of immune cells.

Cellular heterogeneity correlated with clinical pathology of breast cancer

Cellular heterogeneity is an important part of the tumor microenvironment and is necessary for the growth and development of a tumor. We studied whether certain cell types were significantly correlated with certain clinical parameters, including age, sex, T stage, N stage, M stage, and TNM stage. A number of cell types were significantly correlated with clinical parameters, especially T stage and M stage (Fig. 4A). Astrocytes, mesangial cells, and mast cells were negatively correlated with the T stage and plasma cells were positively correlated with the T stage (Fig. 4B–4E). CD4+ Tcm, CD4+ Tem, microvascular (mv) endothelial cells, NKT, and MSC were all significantly higher at the M stage in patients with distal metastasis (Fig. 4F–4J). CLP was significantly higher at the N stage in patients with lymph node metastasis (Fig. 4K). Th1 cells and MSC were both positively correlated with TNM stage (Fig. 4L–4M). The 12 male breast cancer patients studied tended to have higher proportion of CLP and NKT compared with the female breast cancer patients (Fig. 4N–4O).

Figure 4 Involvement of cellular heterogeneity in clinic-pathology of breast cancer.

(A) A number of cell types were significantly correlated with clinical parameters. (B to O) Examples of significant correlations between different cell types and clinical parameters, y-axis represents the fractions of each cell type. *, P < 0.05. **, P < 0.01. ***, P < 0.001.

Prognostic model with survival associated cell types

Emerging evidence suggests that the number of tumor-infiltrating lymphocytes (TILs) of primary tumors consistently predicts favorable outcomes for a number of tumor types, including breast cancer. Therefore, survival analyses were performed to find survival-associated cell types within the tumor microenvironment (Fig. 5A). Immune cells were more strongly associated with overall survival, especially CD4+ and CD8+ T cells (Fig. 5A). Most T cells, including CD8+ T cells, CD8+ Tcm, CD8+ naïve T cells, and CD4 Tem, were favorable prognostic factors. However, high CD4+ naïve T cells were associated with worse overall survival (Fig. 5A–5M). NKT, class switched memory B cells, NK cells, cDC, and pDC were also significantly associated with overall survival (Fig. 5A–5M). A number of stromal cells, including mesangial cells and keratinocytes, were favorable prognostic factors but myocytes were adverse prognostic factors (Fig. 5A–5M). Multivariate COX regression revealed that CD8+ T cells, mesangial cells, keratinocytes, NKT, and class switched memory B cells were independent prognostic factors. We built a prognosis predictor model with five independent prognostic factors. Our model more reliably determined the survival of breast cancer patients with the highest AUC of ROC of 0.708, versus when the factors were analyzed separately (Fig. 5N, 5O).

Figure 5 Survival associated tumor-infiltrating cells in breast cancer.

(A) Forest plot of hazard ratios of survival associated cell types. (B to M) Kaplan-Meier curves of survival associated cell types. Red lines indicated high fraction while blue lines indicated low fraction of each cell types, respectively. (N) Kaplan-Meier curves of predictor built with five independent prognostic factors ( P < 0.01). O. ROC curves of prognostic model and the five independent prognostic factors.

Subtypes of breast cancer had diverse phenotypes of cellular heterogeneity

Emerging evidence suggests that the breast cancer transcriptome has a wide range of intratumoral heterogeneity, as well as genomic heterogeneity based on ER, PR, and Her2, which are determined by the tumor cells and immune cells in the surrounding microenvironment (Chung et al., 2017). We explored cellular heterogeneity among different subtypes of breast cancer by Her2, ER, and PR status. 1,092 breast cancer patients were classified into five groups according the clinicopathological parameters provided by TCGA, including 30 Her2+_HR- patients, 59 Her2+_HR+ patients, 426 Luminal type (Her2-_HR+) patients, 97 triple negative (Tri-negative) patients, and 480 unknown patients (Table 1). The relative proportion of different cells varied widely among these five subtypes (Fig. 6). Tri-negative patients had the highest fraction of immune cells while luminal type patients had the smallest fraction of immune cells, especially CD4+ and CD8+ T cells (Fig. 6A, Fig. S1). Cluster analyses from ImmuneScore, StromaScore, and heatmap suggested the absence of certain cell types used to distinguish these five subtypes (Fig. 6B). Furthermore, t-SNE cluster analysis suggested a large difference in tumor-infiltrating cells among these five subtypes (Fig. 6C and 6D). B cells, T cells, macrophages, Th cells and stromal cells, including keratinocytes, were significantly differentially enriched in these subtypes (Fig. 6E–6P and Fig. S2). Tri-negative breast cancer tissues had the highest fractions of plasma cells, pro B cells, macrophages M1, Th1, and Th2 cells, but M2 cells had the lowest fraction of macrophages (Fig. 6E–6L). Keratinocytes, sebocytes, and pericytes were found frequently in Tri-negative breast cancer whereas MSC cells were found in low amounts (Fig. 6M–6P). Survival analysis revealed interesting differences between the five subtypes (Fig. 7). Each subtype of breast cancer had a unique pattern of survival-associated tumor-infiltrating cells. Different cell types may have different functions in the prognosis of breast cancer patients. Keratinocytes had a favorable effect on the prognostic factors while neurons were associated with adverse prognosis factors in luminal type patients (Fig. 7A). However, in Tri-negative patients, keratinocytes predicted a worse overall survival and neurons predicted a better overall survival (Fig. 7D). Taken together, the diversity of cellular heterogeneity among the different subtypes of breast cancers suggested that tumor-infiltrating cells within the tumor microenvironment were essential in shaping the intratumor heterogeneity of breast cancer.

Figure 6 Differences of cellular heterogeneity among different subtypes of breast cancer.

(A) Median fraction of 64 cell types in five subtypes of breast tumor. (B) Cluster analysis by ImmuneScore and StromaScore, which were calculated by summing up the fractions of immune and stromal cells, respectively. (C) Dimensionality reduction and visualization by t-Distributed Stochastic Neighbor Embedding (t-SNE) clustering with fractions of all the 64 cell types. (D) Heatmap of fractions of 64 cell types in five subtypes of breast cancer. (E to P) Box plots with dots of fractions of certain cell types in five subtypes of breast cancer. *, P < 0.05. **, P < 0.01. ***, P < 0.001, ****, P < 0.0001.

Discussion

We observed distinct tumor-infiltrating cell types within the tumor microenvironment. The abundance and activation status of these cell types is of interest to researchers for their novel bioinformatic techniques. Tumor-infiltrating cells are known to play important roles in the regulation of tumor proliferation, metastasis, and invasion (Galon et al., 2006; Hanahan & Coussens, 2012). The rapid accumulation of high-throughput data and the evolution of bioinformatics algorithms allows us to digitally dissect the interactions between tumors cells and tumor-infiltrating cells, including immune cells and stromal cells (Aran, Hu & Butte, 2017a; Hackl et al., 2016). The high-throughput approach may help understand the complexity of the tumor microenvironment and lead to innovations in breast cancer treatment and prognosis. xCell analysis reveals that plasma cells and CD4+ Tcm in the tumor microenvironment may play a role in the progression of triple-negative breast cancer (Deng et al., 2019), although only immune cells were investigated.

Figure 7 Survival associated tumor-infiltrating cells in five subtypes of breast cancer.

(A to E) Forest plots of hazard ratios of survival associated cell types in five subtypes of breast cancer.

We used the digital deconvolution from xCell to determine the cellular heterogeneity within breast tumor and normal tissues. A total of 64 cell types with more than 30 immune cell types were characterized at high resolution. This was the most studied set of cell types, especially for tumor-infiltrating lymphocytes (TILs). We focused on immune cell types, especially the CD4+/CD8+ T cells, and discovered differences between breast tumor tissues and adjacent normal tissues with polarized enrichment of certain cell types. Our results demonstrated that the expression of inhibitory receptors (including PD1, CTLA4, LAG3, and TIM3) were positively correlated and were associated with certain types of T cells in tumor tissues, especially CD8+ Tcm and CD8+ T cells. CD4+ Tem, CD8+ Tcm, CD8+ T-cells, CD8+ naive T-cells, and B cells were associated with better prognosis whereas CD4+ naive T-cells were negatively associated with prognosis for breast cancer patients. Innate and adaptive immune cells had active immune responses to tumor antigens, including T cells, B cells and DC. TDRD6 and TTK are promising targets for cancer vaccines that could activate a number of immune cells, especially T cells and B cells. Stromal cells were also widely involved in the development of breast cancer. Endothelial cells and fibroblasts were not observed as frequently in tumor tissues. Astrocytes and mesangial cells were negatively correlated with T stage. Mesangial cells and keratinocytes were favorable prognostic factors and myocytes were adverse prognostic factors. We built a prognosis predictor with survival-associated cell types to determine the overall survival of breast cancer patients. Cellular heterogeneity was also profiled in different subtypes of breast cancer based on Her2, ER, and PR status. Five subtypes of breast cancer demonstrated various phenotypes and the cell types may have had different or opposing roles in each subtype of breast cancer.

Immunotherapies, including immune checkpoint blockers, therapeutic vaccines, and engineered T cells are being intensively investigated (Schumacher & Schreiber, 2015) to determine how tumor cells interact with immune cells. The tumor-immune cell interaction poses considerable challenges since the development of cancer and immune surveillance by innate and adaptive immune cells with plasticity and memory are evolving ecosystems. The complex interplay between solid tumors and host immunity has been widely studied but is not well understood. Tumor infiltrating lymphocytes (TILs) have been associated with clinical outcomes in many tumor types (Anagnostou & Brahmer, 2015; Schoenfeld, 2015). For example, CD8+ TILs are prognostically favorable in melanoma, colorectal, ovarian, and non-small cell lung cancer. CD8+ TILs are able to kill tumor cells in specific cancers (Yee et al., 2002). Immunity in breast cancer remains largely unstudied with only a few preliminary evaluations on the prognostic value of CD4+/CD8+ T lymphocytes. The presence of TILs is potentially predictive and prognostic in specific breast cancer subtypes, especially in patients with human epidermal growth factor receptor 2 positive and triple-negative breast cancer. Large adjuvant studies have shown that higher levels of TILs in primary biopsies are associated with improved overall survival and fewer recurrences, regardless of therapy (Adams et al., 2014; Dieci et al., 2015; Loi et al., 2013).

We provided detailed information about the immune cells in breast cancer with numerous novel findings. Inhibitory receptors were expressed on certain types of T cells, preferring CD8+ T cells and CD8+ Tcm. The co-expression of PD1, CTLA4, LAG3, and TIM3 were more commonly observed in tumor tissues compared with normal tissues, which may explain the limited effects of a single immune checkpoint inhibitor and the use of combined strategies. The simultaneous inhibition of PD1 and CTLA4 (Wolchok et al., 2013) or TIM3 (Fourcade et al., 2010) in advanced melanoma patients show promise in clinical trials. CD8+ naive T cells versus CD4+ naive T cells were favorable prognostic factors for the overall survival of breast cancer patients, suggesting that not all T cells were protective. These results suggest that the upregulated co-expression of multiple immune inhibitory receptors may contribute to immune suppression. More attention should be paid to subtypes of T cells when using immune checkpoint blockers since immune cells are highly conditional and may have different or even opposing roles in response to tumor cells.

Growing evidence suggests that immune cells and tumor cell-extrinsic factors, including fibroblasts, endothelial cells, adipocytes within the tumor microenvironment have important roles in inhibiting apoptosis, enabling immune evasion, and promoting proliferation, angiogenesis, invasion, and metastasis (Whiteside, 2008). We found that endothelial cells were significantly higher in adjacent normal tissues (Fig. 1V) and breast cancer patients with metastasis had a higher fraction of microvascular (mv) endothelial cells (Fig. 4H). A high level of mv endothelial cells was significantly associated with worse overall survival (Fig. 5A). Recent studies have shown that endothelial cells may promote triple-negative breast cancer cell metastasis via PAI-1 and CCL5 signaling (Zhang et al., 2018). The presence of endothelial cells significantly enhanced the angiogenic activity of breast cancer cells (Buchanan et al., 2012). These results support our analysis and further study of the clinical relevance of these cell types may provide novel insights into the initiation and progression of breast cancer.

We analyzed and described the potential roles of different tumor-infiltrating cells. Our study would benefit from additional analysis and experimental validations to further investigate the roles of the 64 types of cells profiled in this study.

Conclusions

We revealed the landscape of cellular heterogeneity at high resolution and provided novel insights into cell interactions within the tumor microenvironment in breast cancer. Our results may assist in the development of future therapeutic and predictive strategies. Further study should focus on the subtypes of immune cells and stromal cells identified in this study.

Supplemental Information

Supplemental Information 1 Median fractions of 64 types of cells in five subtypes of breast cancer

Click here for additional data file.

Supplemental Information 2 Diverse differences of 64 types of cells among the five subtypes of breast cancer

Click here for additional data file.

Additional Information and Declarations

Competing Interests

Author Contributions

Data Availability

The authors declare there are no competing interests.

Yuan Li conceived and designed the experiments, performed the experiments, analyzed the data, prepared figures and/or tables, authored or reviewed drafts of the paper, and approved the final draft.

Zuhua Chen performed the experiments, prepared figures and/or tables, authored or reviewed drafts of the paper, and approved the final draft.

Long Wu and Junjie Ye analyzed the data, authored or reviewed drafts of the paper, and approved the final draft.

Weiping Tao conceived and designed the experiments, authored or reviewed drafts of the paper, and approved the final draft.

The following information was supplied regarding data availability:

All raw data are publicly accessible at TCGA using search term TCGA-BRCA.

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
