# Peer review of "Cellular heterogeneity map of diverse immune and stromal phenotypes within breast tumor microenvironment"

_PeerJ, doi:10.7717/peerj.9478_

## Round 0.1 · original submission · Major Revisions

While generally enthusiastic, the reviewers have a number of questions/concerns. Please review and address these comments as part of your revision.

Reviewer 1 ·

Basic reporting

1.1 Discussion should be added to existing relevant reports.
Eg.
[1] Deng L, Lu D, Bai Y, et al. Immune Profiles of Tumor Microenvironment and Clinical Prognosis among Women with Triple-Negative Breast Cancer [J]. Cancer Epidemiol Biomarkers Prev, 2019, 28(12): 1977-1985. DOI: 10.1158/1055-9965.EPI-19-0469.

Experimental design

No comment.

Validity of the findings

No comment.

Additional comments

Breast cancer is one of the predominant types of tumors in women worldwide. Despite great progress in the early diagnosis and treatment recently, drug resistance and distal metastasis remain major causes of cancer-related mortality. Tumor microenvironment (TME) cells are important elements in tumor tissue. There is increasing evidence that they have important clinical pathological significance in predicting tumor clinical outcomes and therapeutic effects. It is inspiriting and beneficial to analyze the global view of tumor-infiltrating immune and stromal cells in tumor tissues and adjacent normal tissues, and investigate the correlation of TME cells’ signature with breast cancer prognosis. In this manuscript, Dr. Li and colleagues looked at the role of the “64 types of immune and stroma cells’ signature” in diagnosis and prognosis of breast cancer. The study’s conclusion showed that "immune and stroma cells’ signature" was expected to be used to predict the prognostic survival of breast cancer. Although the current study is well and interesting, there are some major concerns that need to be addressed before consideration of publication.

Major concerns:
1. In the newly published article [1], Dr. Deng and colleagues also analyzed BRCA dataset from TCGA using xCell, so it is necessary to discuss the similarities and differences between the two studies.
[1] Deng L, Lu D, Bai Y, et al. Immune Profiles of Tumor Microenvironment and Clinical Prognosis among Women with Triple-Negative Breast Cancer [J]. Cancer Epidemiol Biomarkers Prev, 2019, 28(12): 1977-1985. DOI: 10.1158/1055-9965.EPI-19-0469.
2. “Abstract – Methods”. The method should be described more detailed.
3. “Abstract – Results”. The results should be briefly descripted.
4. “Results – Breast tumor tissues had higher fractions of immunes cells than normal tissues” Line 128-142, Figure 1D to J. Paired samples (e.g. 112 cases) are recommended for comparison, in order to reduce the error caused by differential number of samples analyzed between groups.
5. “Results – Inhibitory receptors were co-expressed on certain subtypes of T cells” Line 150-154, Figure 2A and B. In order to show similarities and differences between tumor samples and normal tissues, the Tumor (F2A) and normal tissues (F2B) should be combined on a heat map, and t-Distributed Stochastic Neighbor Embedding (t-SNE) analysis should be preform.
6. “Results – Prognostic model was built with survival associated cell types” Line 211-214, What is the meaning of “…prediction model…”? The method should be described more detailed, including software, risk factor calculation formula, and references.
7. “Results – Subtypes of breast cancer had diverse phenotypes of cellular heterogeneity” Line 229. What is the meaning of “…ImmuneScore and StromalScore…”? The method should be described more detailed.

Minor comments:
1. “Figure 1B”. The information of the axis and unit should be added to depict the “blue to red” color.
2. “Figure 1D to J”. Missing the y-axis label.
3. “Figure 2A to D”. The information of the axis and unit should be added to depict the “blue to red” color.
4. “Figure 2E to H”. The scatter plot should show P-values and R-values.
5. “Figure 3B to F”. Missing the y-axis label.
6. “Figure 4 figure legend”. “ESCC”? Please examine your manuscript carefully.
7. “Figure 5 B to N”. The values of cutoff (can be noted “best”), Hazard Ratio (HR), and 95% Confidence Interval (CI) of ratio should be shown in survival curve, and the unit of survival time should use months or years.
8. “Results – Subtypes of breast cancer had diverse phenotypes of cellular heterogeneity” Line 231-232. “…t-SNE cluster analysis…” What data analyzed? Is RNA profile or the fractions for 64 cell type? Please state clearly.
9. “Results – Subtypes of breast cancer had diverse phenotypes of cellular heterogeneity” Line 231-232. Did “…distinguish these five subtypes (Figure 6B-C)…” should be “ (Figure 6B)”; and “…among these five subtypes (Figure 6D)…” should be “(Figure 6C-D)”?
10. “Figure 6D to F”. The text in charts is too small to read, so charts need to be rearranged.
11. “Figure 7”. The text in charts should not be distorted and should be easily recognizable to the reader.

Reviewer 2 ·

Basic reporting

no comment

Experimental design

no comment

Validity of the findings

no comment

Additional comments

In the current study, the authors aimed to provide a unique cellular heterogeneity map of diverse immune and stromal phenotypes in breast cancer microenvironment and uncover potential therapeutic targets and biomarkers. xCell was used to portrayed the cellular heterogeneity map of 64 tumor microenvironment -associated cell types in RNA-seq dataset of breast cancer and normal tissues downloaded from TCGA. They found that specific immune and stromal cells, such as CD8+ T cells, keratinocytes, NKT and Class switched memory B cells in the tumor microenvironment, may serve as independent prognostic factors in the subsequent progression of breast cancer. Furthermore, they analyzed a ImmuneScore and StromalScore system to distinguish the five subtypes of breast cancer, and they owned diverse phenotypes of cellular heterogeneity. The research is interesting. I have some major concerns as the following:
1. The method used in this study should be described in more details, and a flowchart of study design and patient’s information should be added.
2. How did the authors calculate the ImmuneScore and StromalScore from 64 cell types? And the scores of each immune and stromal cell should be demonstrated in table.
3. What did the prognostic prediction model exactly mean in Figure 5? The evalution criteria of significant prognosis factors used to predict the survival should be provided.
4. It would be easier to understand if the illustration of figure legend was presented in more specifically way.
5. As shown in Figure 2E-H, the immune inhibitory receptors were significantly correlated with certain subsets of T cells. The correlation coefficient and P value should be provided in scatter plot.
6. What is the y-axis supposed to be in Figure 1D-J, Figure 4B-F and Figure 6E-F?
7. The figure legend title in Figure 4 “Involvement of cellular heterogeneity in clinic-pathology of ESCC” was miswritten, it should be “breast cancer”.
8. The number of samples of each group should be provided in figure.
9. Writing of this article may need further modification.

Reviewer 3 ·

Basic reporting

The manuscript is generally well written, but there are some issues with English usage that will need to be addressed. This might be accomplished by a native English speaker upon revision by the authors, or by the journal's scientific editor and production staff.

The literature cited is appropriate and complete.

The manuscript is nicely prepared. The figures look good. Table 1 describes the cell types evaluated in this study and provides abbreviations when applicable. This table is important for its content, but the content might be better delivered in the text rather than a table.

**It is not at all clear that the raw data is available for other investigators. This study is based upon 1092 patients from the TCGA. There is no description of how patients were selected and so whereas the TCGA data is available, it is not possible to reconstruct this cohort.

Experimental design

This manuscript describes a study of 1092 breast cancers from TCGA where the gene expression signatures were used to identify 64 cell types including immune cells, stromal cells, and other cell types. Given that the cellular composition of a cancer can impact on its biology and by extension its treatability and prognosis, this is considered a significant study. Overall, the authors used sound methods and applied good statistical analysis.

1. The authors curated gene expression data from TCGA for 1092 breast cancers. No selection criteria are provided and so it is not possible for a reader to recreate this data set. It is not clear that the raw data is shared in the form it was used.
2. The authors don't provide summary statistics for the breast cancers examined, such as age of patients, ER/PR/HER2 status, etc. This information is contained in the results, but really belongs in the Methods. In fact, the cohort included 12 make breast cancers and this was not revealed until late in the results.
3. Some of the text included in Results (like the first paragraph on cell types analyzed) could be included in the Methods.

Validity of the findings

This manuscript describes an interesting study. The authors used good methods and statistics. The figures are nicely prepared. The text of the results and discussion is well done. Overall, the findings are very interesting and presented in a fairly straightforward manner. In general, the approach taken by these investigators provides a nice discovery study. What it lacks is a validation study where a smaller subset of breast cancers is analyzed in a different manner (immunostaining for instance) to verify the gene expression mining results.

1. The results shown in Figure 1 include 112 normal breast tissues in comparison to 1092 breast cancers. The authors have chosen to show results for all cancers compared to the normals instead of using the paired tissues in a much more powerful manner.
2. Figure 4 includes "ESCC" in the figure title. Please define this abbreviation. I would not expect this abbreviation in a breast cancer manuscript.

Additional comments

Overall, this is a very well prepared manuscript that describes an interesting study. The manuscript can be improved through minor edits, and perhaps consideration of additional analysis (paired samples) and/or a validation study (with immunostaining of tissues anlyzed by gene expression).

---

## Round 0.2 · accepted · Accept

Thank you for addressing the reviewers' concerns. Do check on the figure comments brought up by reviewer 1, though these issues may be taken care of during the formatting of the manuscript for publication. Congratulations again.

Reviewer 1 ·

Basic reporting

No comment.

Experimental design

No comment.

Validity of the findings

No comment.

Additional comments

Breast cancer is one of the predominant types of tumors in women worldwide. Despite great progress in the early diagnosis and treatment recently, drug resistance and distal metastasis remain major causes of cancer-related mortality. Tumor microenvironment (TME) cells are important elements in tumor tissue. There is increasing evidence that they have important clinical pathological significance in predicting tumor clinical outcomes and therapeutic effects. It is inspiriting and beneficial to analyze the global view of tumor-infiltrating immune and stromal cells in tumor tissues and adjacent normal tissues, and investigate the correlation of TME cells’ signature with breast cancer prognosis. In this manuscript, Dr. Li and colleagues looked at the role of the “64 types of immune and stroma cells’ signature” in diagnosis and prognosis of breast cancer. The study’s conclusion showed that "immune and stroma cells’ signature" was expected to be used to predict the prognostic survival of breast cancer. The paper is improved and most concerned raised by the reviewer have been addressed. I think it is might suitable for publication at this version of revised manuscript.

Minor comments:
1. Figure 4 B and C. The two side-by-side charts are not the same size. To maximize the quality of the article, set the same type of chart to the same magnification as possible.
2. Figure 4 K to O. Pictures in the same row, be sure to align horizontally. Authors should check all the pictures one last time.

Reviewer 2 ·

Basic reporting

No comment

Experimental design

No comment

Validity of the findings

No comment

Additional comments

In the revised manuscript, the authors provided a unique cellular heterogeneity map of diverse immune and stromal phenotypes in breast cancer microenvironment using xCell method, which might uncover potential therapeutic targets and biomarkers. They found that specific immune and stromal cells in the tumor microenvironment may serve as independent prognostic factors in the subsequent progression of breast cancer. The authors have revised the paper carefully according to the concerns raised by reviewers in the current version. I think it is suitable for publication at this point for this version of revised manuscript.